# Evolution of Microstructure in Friction Stir Processed Dissimilar CuZn37/AA5056 Stir Zone

**DOI:** 10.3390/ma14185208

**Published:** 2021-09-10

**Authors:** Anna Zykova, Andrey Chumaevskii, Anastasia Gusarova, Denis Gurianov, Tatiana Kalashnikova, Nickolai Savchenko, Evgeny Kolubaev, Sergei Tarasov

**Affiliations:** Laboratory of Local Metallurgy in Additive Technologies, Institute of Strength Physics and Materials Science Siberian Branch of Russian Academy of Sciences, Akademicheskiy pr., 2/4, 634055 Tomsk, Russia; zykovaap@mail.ru (A.Z.); tch7av@gmail.com (A.C.); gusarova@ispms.ru (A.G.); desa-93@mail.ru (D.G.); gelombang@ispms.tsc.ru (T.K.); savnick@ispms.tsc.ru (N.S.); eak@ispms.ru (E.K.)

**Keywords:** friction stir processing, dissimilar joint, intermetallic compounds, normal fracture

## Abstract

Dissimilar friction stir processing on CuZn37/AA5056 was performed to study structural and phase evolution of a friction stir zone. Formation of 5–10 μm intermetallic compounds (IMCs) such as Al_2_Cu was the main type of diffusion reaction between copper and aluminum. Other alloying elements such as Mg and Zn were forced out of the forming Al_2_Cu grains and dissolved in the melt formed due to exothermic effect of the Al_2_Cu formation. When solidified, these Zn-enriched zones were represented by α-Al+Al_2_Cu+Zn phases or α-Al+Al_2_Cu+Zn+MgZn regions. Eutectic Zn+MgZn was undoubtedly formed the melt after stirring had stopped. These zones were proven to be weak ones with respect to pull-off test since MgZn was detected on the fracture surface. Tensile strength of the stirred zone metal was achieved at the level of that of AA5056.

## 1. Introduction

Friction stir processing/welding (FSP/FSW) allows for the obtaining of dissimilar metal joints when two different metals or alloys are plasticized and intermixed in the stir zone (SZ). Intermixing usually occurs in a fine-grained solid state [1,2,3,4,5,6,7,8,9,10], but if the alloy components are capable of forming compounds, they may contact and exothermically react to each other when forming an intermetallic compound and thus produce a liquid phase. One of the well-known such systems is Cu–Al, whose friction stir welding (processing) results in forming CuAl IMCs in the stir zone [11,12,13,14,15]. Brittle IMCs are the main problem in dissimilar joints because they reduce electric and heat conductivities as well as spoil mechanical strength of the joint so that various techniques are applied to get rid of them or at least reduce their grain size and content. First of all, these could be minimization of frictional heat generation during the FSP and the use of additional cooling. Moreover, there are data that confirm reduction of IMC content after using in situ sonication [16] and intercalation of Zn-layer between Al and Cu with a formation of a thin IMC diffusion barrier [17]. An attempt was undertaken by Avettand-Fènoëla et al. to study the effect of Zn on linear friction welding Al and Cu to brass [18]. Melting of a zinc filler due to its low thermal diffusivity as well as formation of dendritic structures was observed. Alaeibehmand et al. inserted a zinc foil between steel and aluminum plates but found that double pass friction stir spot welding had greater effect of the joint strength [19]. Brass interlayers were used by Swarnkar et al. to enhance a FSW AA 6082 joint strength [20]. The use of zinc interlayer for FSW lap joining AZ31B/7075-T6 alloys resulted in obtaining modified microstructures with fine Mg–Zn IMCs distributed in the stir zone [21].

These and many other efforts were focused on creating a transition (barrier) layer between dissimilar metals to limit formation of detrimental IMCs. However, there is an alternative approach to use multipass friction stir processing for obtaining composite materials with a metal matrix reinforced by in situ-formed IMC particles [22]. Let us note that in fact FSW and FSP on dissimilar metals share the same structural evolution processes and therefore can be described using the same terms. In this connection, it is desirable to obtaining fine IMC particles evenly distributed in a stir zone, i.e., effective stirring is desirable, which is often prevented by strong adhesion between the stirred material and FSW tool [23]. Introduction of a low-melting element such as zinc would improve the intermixing and distribution of IMC. When forming an in situ composite zone and hoping to obtain a homogeneous IMC distribution, it would be more reasonable to introduce this brazing element not as a foil or a zinc powder layer but as a component of an alloy such as brass.

CuZn brass possesses heat and electric conductivities comparable to those of copper, as well as much higher mechanical strength and corrosion resistance. These characteristics call for using it, for example, in heat-exchanging devices as a substitute for copper. Dissimilar brass/aluminum alloy joints are also promising because of reduced costs.

IMCs formed in such a joint include both Al/Cu- and Zn-based ones, as shown by [4,5,24,25,26,27]. Some investigations showed formation of hook-like defect structures on the Al/brass interface around the SZ that acted like a stress concentrator [4,5,26]. Formation of such a defect is common with the lap FSW. The Al_x_Cu_y_ IMCs formed in the vicinity of the joint line were the reason behind low dissimilar joint strength.

Brittle Al_2_Cu, Al_4_Cu_9_, and CuZn IMCs were found in the SZ in the lap welded AA5083/CuZn34 joint [26]. The IMC layer thickness was a function of FSW tool rotation rate and welding speed. Moreover, defects were obtained when welding at low tool rotation rate. Intermetallic compounds such as Al_4_Cu_9_, AlCu, CuZn, Cu_4_Zn, CuZn_5_, Al_2_Cu, and Al_4.2_Cu_3.2_Zn_0.7_ were detected after friction stir brazing of AA6061 and H62 brass [24]. This IMC layer thickness proved to be dependent of the welding speed, and thus increasing the welding speed from 20 to 60 mm/min resulted in reducing the IMC thickness from 60 to 16 μm. Further increasing the welding speed led to crack formation.

These examples testify that IMCs obtained in the stir zone between aluminum alloy and brass involve all main components of both alloys. The roles played by main components of dissimilar joint alloys such as Al, Cu, and Ti in forming the in situ IMCs are extensively studied. However, all alloys contain numerous other alloying elements that also could react during FSP and thus have effect on the formation of SZ according to solid-state diffusion or solidification from a melt.

This issue is very important, but not much attention has been given to it thus far. The objective of this research is, therefore, to study the microstructural evolution of SZ in dissimilar lap FSP (FSW) of a Cu–Zn alloy to an Al–Mg alloy, with special attention given to formation of primary Al/Cu and secondary Mg/Zn IMCs.

## 2. Materials and Methods

Sandwich bimetallic workpieces for FSP were prepared by placing and fixing 200 × 60 × 2 mm^3^ sheets of CuZn37 on the AA5056 aluminum alloy 200 × 60 × 5 mm^3^ ones (Figure 1). Element compositions of the metals are shown in Table 1. Ultimate tensile strength levels of AA5056 and CuZn37 at T = 20 °C were ≈290 MPa and ≈340 MPa, respectively.

A pilot FSP machine located at the facilities of Institute of Strength Physics and Materials Science Siberian Branch of Russian Academy of Sciences (ZAO Sespel, Cheboksary, Russia) was used for FSP (lap FSW) on these samples using a three-flute truncated cone tool with 2.5 mm height pin and ∅12 mm shoulders mounted at an angle of 3° with respect to normal. The FSP parameters were used as shown in Table 2. These parameters were chosen as optimal ones on the basis of our previous experiments. Two FSP passes were applied to improve the homogeneity of the SZ as well as avoid defects. Higher rotation rate during the second FSP pass was intended to improve plasticization of the IMCs formed during the first pass.

Samples for microstructural and mechanical characterization were cut off the FSPed tracks using a wire-cut electric discharge machine DK7745 (Suzhou Simos CNC Technology Co., Ltd. Suzhou, China) (Figure 1). Metallographic microscope Olympus LEXT 4100 (Olympus NDT, Inc., Waltham, MA, USA) and scanning electron microscope (SEM) Zeiss LEO EVO 50 (Carl Zeiss, Oberkochen, Germany) were used for microstructural characterization of the stir zone (SZ) transversal and longitudinal section views obtained after FSP. Chemical element distributions were examined using an energy-dispersive X-ray spectroscopy (EDS) attachment to the above-indicated scanning electron microscope. Thin foils for transmission electron microscope (TEM) JEOL-2100 (JEOL Ltd., Akishima, Japan) were cut out of the SZ, mechanically ground, polished, and ion-thinned (Figure 1). Phase composition of the SZ was determined using an X-ray diffractometer XRD-7000S (Shimadzu, Kyoto, Japan), $Co_{K_{\alpha }}$ radiation. Tensile tests were carried out using a tensile machine UTS-110M (TestSystems, Ivanovo, Russia) and samples oriented with their tensile axes along the FSP track length (Figure 1). Pull-off tests were intended to determine the strength of bonding between the lap welded sheets using the cross-shaped samples (Figure 1).

Microhardness tester Duramin 5 (Struers A/S, Ballerup, Denmark) operated at 490.6 mN (50 g) load, and indenter dwell time of 10 s was used for obtaining the microhardness profiles across the SZ.

## 3. Results

### 3.1. Longitudinal Sectioning of AA5056/CuZn37 Stir Zone

Optical macrographs of structures produced by single-pass FSP on AA5056/CuZn37 show that a discontinuity was formed initially as oriented along the FSP track direction (Figure 2a). Alternating pattern composed of either aluminum and brass alloy layers as well as intermetallic compounds (IMCs) were formed in the bottom part of the track during further FSP (Figure 2a, enlarged views of fragments 1 and 2).

The double pass FSP eliminated the discontinuity in the onset part of the track and provided better mechanical intermixing of the metals and formation of solid solution (SS) and IMCs (Figure 2b). Only 2-pass more homogeneous FSPed samples were then characterized for microstructure and strength.

SEM BSE macrographs of the bottom SZ section as viewed along the FSP track length showed layered inhomogeneous structures containing components of different brightness levels on the basis of the atomic numbers of their constituent elements (Figure 3a,d). Bright finger areas in Figure 3a,b,d,e are CuZn37 alloy layers formed by FSP metal flow pattern, while the dark ones are AA5056. Transition reaction zones were between the CuZn37 and AA5056 areas. These transition zones may have been of two types, as shown in Figure 3b,e. Transition zone in Figure 3b shows faceted grains and eutectic structures that could be evidence of solidification from a melted state.

The EDS profiles (Figure 3c,f) obtained along the yellow lines shown in Figure 3a,e allowed for the observation of the alloy element distributions across the SZ. According to the EDS profiles, there were components that can be identified as follows: (1) dark solid solution of Zn (≤12 atom %) and Mg (≤5 atom %) in α-Al, i.e., α-Al (Mg, Zn); (2) IMC Al_2_Cu, Zn (≤9 atom %) and Mg (≤2 atom %); (3) metastable IMC MgZn. The end part of the profile in Figure 3f shows unreacted CuZn37 (Figure 3e).

The α-Al (Cu, Zn) grains appeared as dark, irregular-shaped ≈5–10 μm grains (Figure 3a,b,e and Figure 4), whereas Al_2_Cu IMCs were gray formations with their shapes depending on the their locations. Tetragonal faceted Al_2_Cu were found in regions solidified directly from the melted state (Figure 3b and Figure 4), while irregularly shaped ones were found in regions subjected to stirring and intermixing (Figure 3e).

EDS element mapping in Figure 4a allows for the observation that bright regions correspond to those enriched with Mg and especially with Zn, which can be identified as Mg–Zn solid solution. Eutectic (dendritic) structures in Figure 3b and Figure 4b could be also formed only from the liquid state. The eutectic regions contain mainly Mg and Zn, with Zn concentrations slightly lower as compared to neighboring bright areas without the eutectics. Considering concentrations of Mg and Zn in Figure 4b, we can suggest that the eutectics could be MgZn precipitates in a Zn-rich matrix. Such a conclusion will be confirmed later by XRD from the fracture surfaces, which showed that the precipitates were MgZn IMCs (see Figure 12).

Exothermic diffusion reaction between copper and aluminum results in formation of IMCs, while both Mg and Zn can be forced out of the reaction zone into free spaces where they can form the MgZn. Total content of Mg in the eutectic regions achieved 23.7 atom %, i.e., almost the same value as that of zinc (27.2 atom %) (Figure 4b). Let us note here that element concentrations indicated in Figure 4 EDS maps relate to total Figure 4a,b areas. Therefore, concentration of aluminum in the eutectic regions as high as 42.2 atom % is mainly due to residual aluminum alloy grains at the periphery, while inside the eutectic region it is negligible.

### 3.2. Transversal Sectioning 

The stirring zone in plane transversal to the FSP track direction is represented by layered structures obtained by transfer and intermixing the aluminum alloy with brass (Figure 5a). These layered structures are composed of oversaturated solid solutions (Figure 5c–e, pos. 1) and IMCs (Figure 5a–e, pos. 2). According to XRD (Figure 5b), the stir zone contain phases as follows: α-Al (Mg, Zn), FCC α-Cu_0.64_Zn_0.36_ solid solution of zinc in copper with a = 3.69612 Å, cubic β-CuZn with a = 2.95000 Å, and θ-Al_2_Cu IMC. Cubic β-CuZn could be formed due to enriching by zinc.

The bottom part of SZ reveals the microstructures similar to that represented in Figure 3e with CuZn areas enriched with 5–9 atom % of Al (Figure 6e, zone I), transition zone II containing dark grains of solid solution of Cu and Zn in α-Al, and gray Al_2_Cu IMCs and Zn precipitates on the IMC grain boundaries (Figure 6c, zone II). Zone III is an α-Al enriched with Zn as a diffusion reaction product (Figure 6c,d). The III zone thickness is about 8 μm. Zone IV is simply an unreacted AA5056.

EDS mapping (Figure 6e–j) shows that transition zone II was enriched with Zn but only small concentrations of Mg, whose distribution did not coincide with that of Zn, i.e., no MgZn IMC formation was suggested in this zone.

TEM image of structures in area 3 (Figure 5d) shows the presence of α-Al and θ-Al_2_Cu IMC grains with fine precipitates identified as Zn, Cu, and CuZn (Figure 7a,b,d and Figure 8a,b,d). CuZn precipitates were clearly observed with the dark field image (Figure 7d) obtained using reflection (–110)_CuZn_. These precipitates look as located within the IMC grains.

On the other side, the θ-Al_2_Cu grains showed the presence of both dislocations and bend contours, which may have been evidence of accommodation deformation in constraint conditions (Figure 8a,c). Dark-field image (Figure 8d) obtained using reflections (002)_Zn_ and (110)_CuZn_ (Figure 8b) allowed for clear identification of free CuZn precipitates, mainly in the IMC areas.

α-Al(Mg,Zn) grains in Figure 8a and Figure 9a show fine rod-like oriented precipitates that could be identified as MgZn_2_. Concentration of zinc in the α-Al(Mg,Zn) grains may have been as high as 2 atom % (Figure 9f), and in fact these grains were close to those of 7000-series alloy.

Globular 0.1–0.5 μm precipitates were found both in the IMC grains and on their boundaries (Figure 9a–g), which were identified as Al–Mn–Fe and Al–Mg–Fe–Zn particles (Figure 9b–f,h–m). The presence of iron was provided by the fact that it was a residual element of both alloys (Table 1). The iron-containing coarse particles are always present with aluminum alloys, and they never dissolve in them during heating below melting. However, in FSW, they may experience strain dissolution and then precipitate again [28].

Fine recrystallized CuZn grains were found in the middle of the SZ on the background of coarse Al_2_Cu IMC grains (Figure 10a,b). These grains may be unreacted CuZn37 grains subjected to plasticization, enriched with Zn, and then transferred to the weld formation zone where they recrystallized in the form of β-CuZn grains.

### 3.3. Mechanical Strength and Microhardness

Normal fracture strength of the FSPed stir zone (lap welded dissimilar CuZn37/AA5056 joint) was tested using a loading scheme as shown in Figure 1. Corresponding stress–strain diagrams in Figure 11 show that the adhesion strength varied as depending on the SZ joint quality. The presence of weak zones and defects was detrimental for the pull-off strength, while eutectic–free showed their strength twice as much as those with defects (Figure 11, curve 1).

Tensile tests demonstrated relatively low ultimate strength with at least three fracture stages (Figure 11b). The first and second stages (Figure 11b, pos. 3 and pos. 4) corresponded to fracture of the SZ and AA5056 substrate. It can be observed here that SZ tensile strength was almost equal to that of AA5056 substrate. Let us note that the standard tensile strength level of AA5056 at T = 20 °C was at the level of 290 MPa.

The fracture surfaces obtained after adhesion tests were examined using the XRD on both sides of the joint. IMC θ-Al_2_Cu(Zn) phase dominated in SZ on both AA5056 and CuZn37 sides (Figure 12a). Moreover, the presence of metastable MgZn, α-Al, α-Cu_0.64_Zn_0.36_, and β-CuZn phases was detected on the CuZn37 side. Similar phases were detected on the AA5056 side (Figure 12b).

Fracture surfaces after normal fracture test allowed for the observation that the pull-off occurred below the CuZn plate in the bottom part of SZ so that IMCs (Figure 13, pos. 3), α-Al (Figure 13, pos. 1) and small CuZn areas (Figure 13, pos. 2) can be observed. It is plausible that CuZn areas were micronecks (dimples) and were the least likely to experience fracture.

The presence of MgZn on the fracture surfaces is a fact that allows for the suggestion that the fracture started and propagated in Zn+MgZn eutectic containing regions, which may be called the weak zones.

Tensile fracture surface demonstrated the presence of the same phases with the top area of unreacted CuZn (Figure 13d, pos. 4) and pull-off zone (Figure 13d, pos. 1).

Microhardness profiles were obtained in three horizontal and three vertical directions with inter-profile distances of 0.9 mm and 1 mm, respectively (Figure 14a). The distance between indentations was 0.1 mm, load 50 g, dwell time 10 s. Microhardness number distribution along lines 1, 2, and 3 represent mechanical responses from shoulder-driven zone, middle pin-driven SZ parts, and bottom pin-driven SZ parts, respectively. The shoulder-driven part contained a dark band where microhardness numbers were high because of formed IMCs (Figure 14b). It can be observed from Figure 14b that IMC regions were mostly located at the SZ sides, while in the bottom SZ part, IMCs were concentrated in the central pin-driven zone (Figure 14d).

Vertical microhardness profiles were obtained across the SZ, starting from the top to the bottom in order to demonstrate inhomogeneity of the IMC distribution. Profile 4 demonstrates that IMCs were formed at the AS in the top and bottom SZ parts (Figure 14e). A similar type of microhardness number distribution was obtained along the centerline 5 (Figure 14f). More compact distribution of IMCs was achieved at the RS side (Figure 14g).

## 4. Discussion

It could be inferred from the metallographic observations that CuZn37/AA5056 stir zone was inhomogeneous by phase composition, containing intermixed layers of unreacted metals with transition layers formed by diffusion reaction between components of CuZn37 and AA5056 (Figure 2). The main interaction was between copper and aluminum, with formation of fine copper/aluminum IMCs (Figure 3). Exothermic effect of this reaction resulted in contact melting and forcing out Zn and Mg from the reaction zone into interparticle spaces.

Depending upon the element concentrations, different phases may form in these spaces after solidification and cooling. The top part of the SZ contained more CuZn and less Al–Mg. Such a situation means that excess quantities of Cu and Zn are left after formation of IMCs. These quantities may partially dissolve in IMCs or stay free to form β-CuZn fine grains alloyed with Fe and Mn when transferred to the zone behind the FSP tool. In addition, this part of the SZ did not contain Z+MgZn eutectics because of low Mg concentration. Instead, the IMCs and residual aluminum alloy grains enriched with Zn were embedded in a Mg–Zn solid solution matrix. The microstructure was in fact a composite consisting of 5–10 Al_2_Cu interlayer. The IMCs formed here were 5–10 μm in size, and such a refining can be related to the grain refining effect of Zn [29].

The bottom part of the SZ contained less copper as compared to that of the top part, and therefore no excess copper was left after IMC formation, while all zinc was forced out of the diffusion reaction zone as well as Mg. These reaction products may form Zn+MgZn eutectics on the IMC boundaries. Formation of Zn+Mg_2_Zn_11_+MgZn_2_ eutectics was observed in an as-cast Zn-1.6 wt. % Mg biodegradable alloy [29]. The fact that CuZn37/AA5056 stir zone contained Zn+MgZn eutectics may be explained by higher content of Mg in the reaction zone (Figure 4b).

Some of the SZ bottom regions were undoubtedly formed by solidification from the liquid state. The fact that undistorted eutectic structures retained after FSP were found in the SZ means that they were formed in a stagnant FSP zone behind the FSP tool as a result of local overheating. Zinc is a low-melting metal (419.5 °C), and therefore Zn-enriched zones may solidify after all others with formation of Zn+MgZn eutectics (Figure 4b).

High content of MgZn IMCs was detected on the fracture surfaces after the pull-off test. It may mean that fracture occurred in the bottom part of the SZ, which contained layers of Zn+(Al-Mg)Zn eutectics. Such a layer may be a weak zone that is responsible for easy fracture, and its formation should be avoided by extra cooling, sonication, or FSP geometry optimization as reported [30].

Eutectic-free zones (Figure 4a) appeared in the transition layers, which were subjected to intense stirring so that only small Zn-enriched zones formed there after cooling. It is suggested that these zones are less prone to fracture as compared to those with eutectics.

Adhesion-assisted transfer mechanism was proposed earlier for FSW [23], being potentially used to explain structural evolution of the CuZn37/AA5056 SZ. Adhesion of plasticized metal to the FSP/FSW tool plays the main role in layer-by-layer formation of the FSW weld joint. Existence of the liquid phase will undoubtedly reduce this adhesion and enhance the efficiency of the SZ metal stirring.

Future efforts will be focused on optimizing FSP conditions with an aim to reduce content of Zn+MgZn eutectics. It can be achieved by using brasses with lower Zn content, ultrasonic-assisted FSP [28], and extra cooling. Control of temperature and welding force during FSP will be necessary in such a study.

## 5. Conclusions

Microstructures and phases formed in the stir zone by dissimilar CuZn37/AA5056 friction stir welding (processing) were studied. The results obtained from such a research allowed for the following conclusions:Formation of Al_2_Cu IMCs from aluminum alloy and brass grains was accompanied by exothermic heating and melting, as well as expelling of both Mg and Zn into intergrain spaces.In the top part of the SZ, Mg and Zn formed a solid solution matrix for 5–10 μm faceted IMCs and irregular-shaped 20–30 μm α-Al (Mg, Zn) grains with fine MgZn_2_ precipitates.The bottom part of the SZ contained irregular shaped grains of Al_2_Cu IMCs and α-Al (Mg, Zn), as well as coarse grains of Zn+MgZn eutectics.Since the pull-off test fracture surfaces were enriched with Al_2_Cu, Zn, and MgZn, it was suggested that these “weak” zones were detrimental for the SZ strength.Tensile strength of the FSPed zone was equal to that of AA5056 substrate.

## Figures and Tables

**Figure 1 materials-14-05208-f001:**
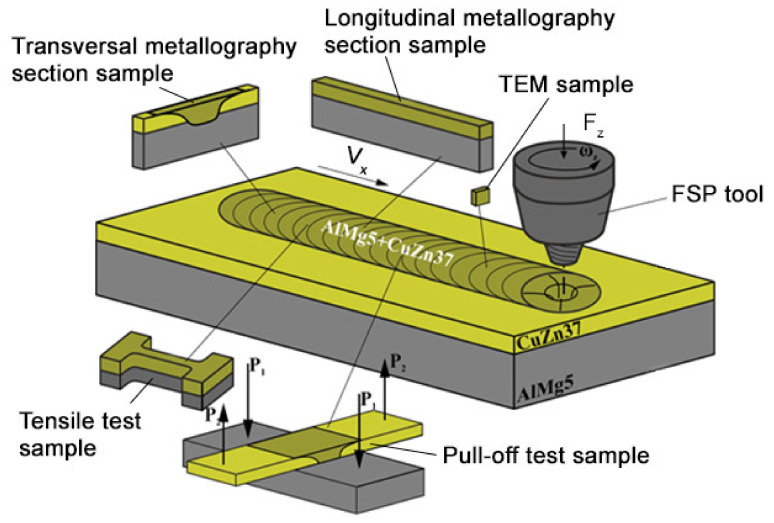
Schematic diagram of FSP on dissimilar metals and sampling.

**Figure 2 materials-14-05208-f002:**
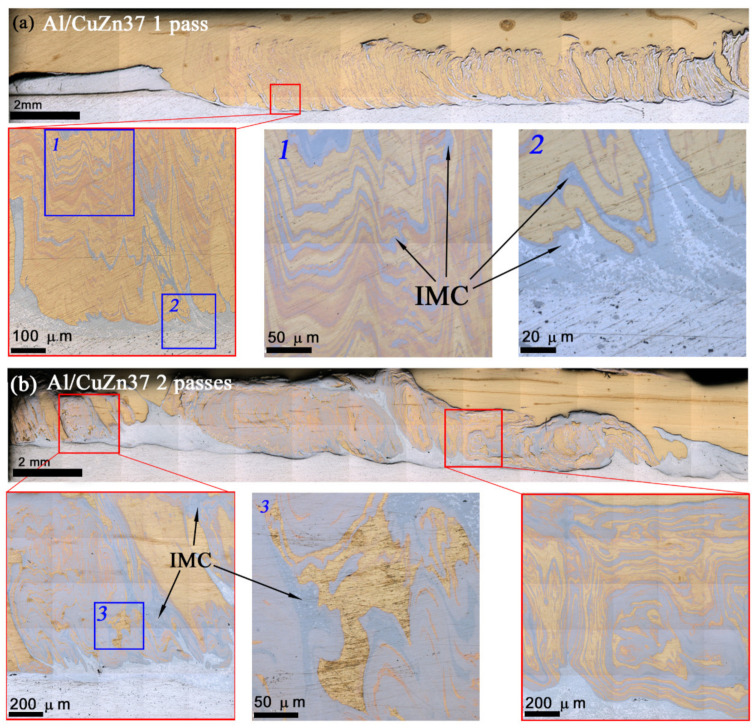
Optical macrographs of AA5056/CuZn37 stir zone obtained after single- (**a**) and double (**b**)-pass FSP as viewed along the FSP track length.

**Figure 3 materials-14-05208-f003:**
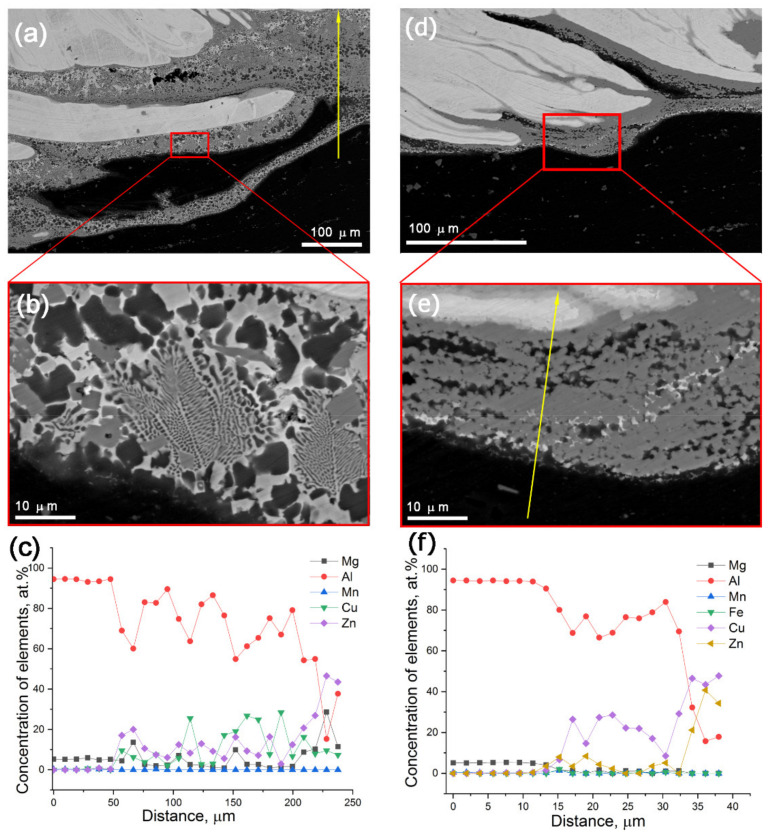
SEM BSE images of areas in the bottom part of SZ as viewed along the 2-pass FSP track length (**a**,**b**,**d**,**e**) and EDS element profiles (**c**,**f**) along the lines in Figure 3a–e.

**Figure 4 materials-14-05208-f004:**
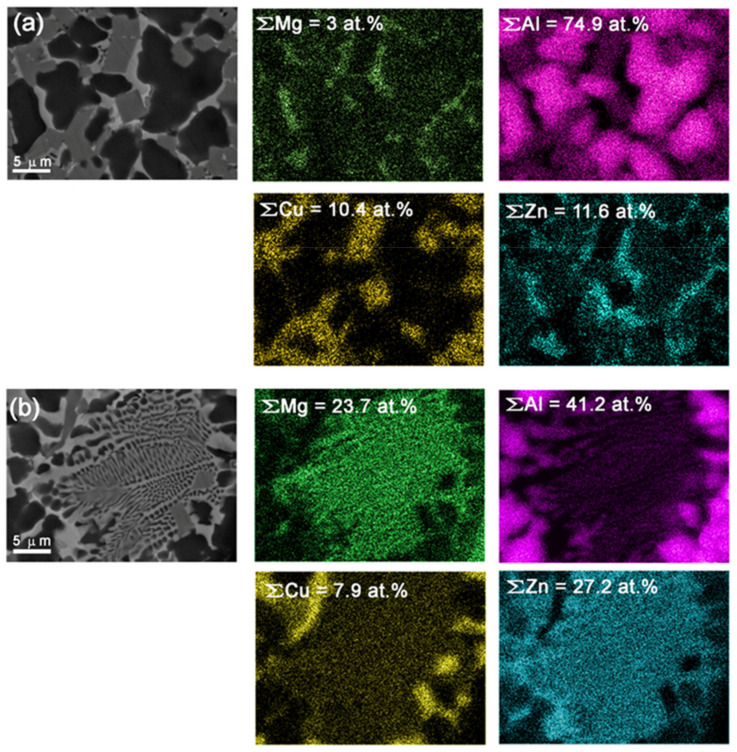
SEM BSE images of α-Al(Cu,Zn), Al_2_Cu IMCs in a Mg-Zn solid solution matrix (**a**), and Zn+MgZn eutectics (**b**) in the bottom part of SZ with corresponding EDS element distribution maps.

**Figure 5 materials-14-05208-f005:**
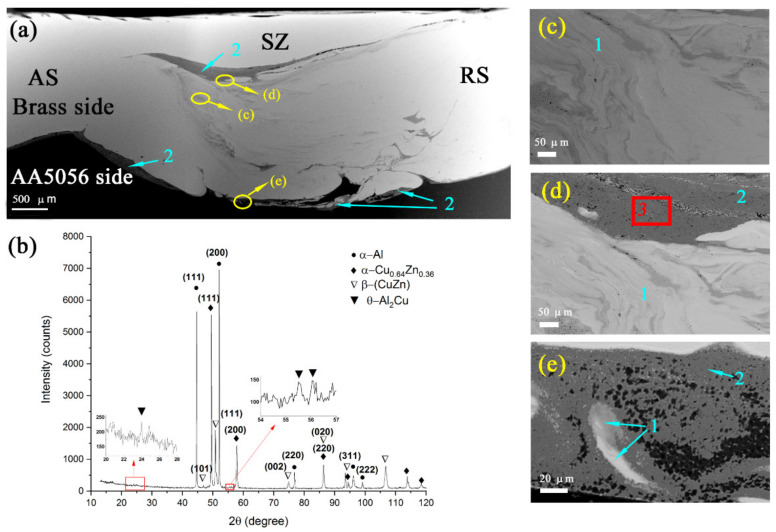
SEM BSE image of AA5056/CuZn37 SZ (**a**), XRD diffractogram (**b**), and enlarged fragments of SZ showing its inhomogeneity (**c**–**e**). 1—region of solid solution, 2—region of IMCs, 3—region where TEM studies were conducted.

**Figure 6 materials-14-05208-f006:**
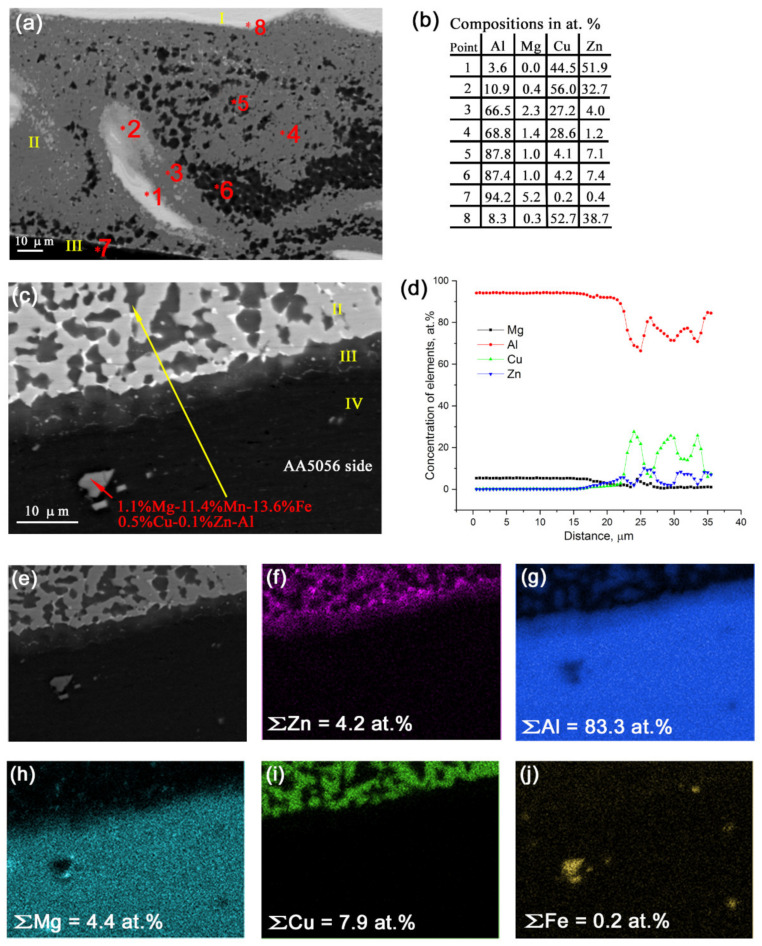
Different regions found in the bottom part of the SZ (**a**,**c**), EDS element concentrations in zones 1 to 8 in Figure 6a (**b**) and EDS profiles along the line in Figure 6c (**d**); EDS element maps (**f**–**j**) obtained from image (**e**).

**Figure 7 materials-14-05208-f007:**
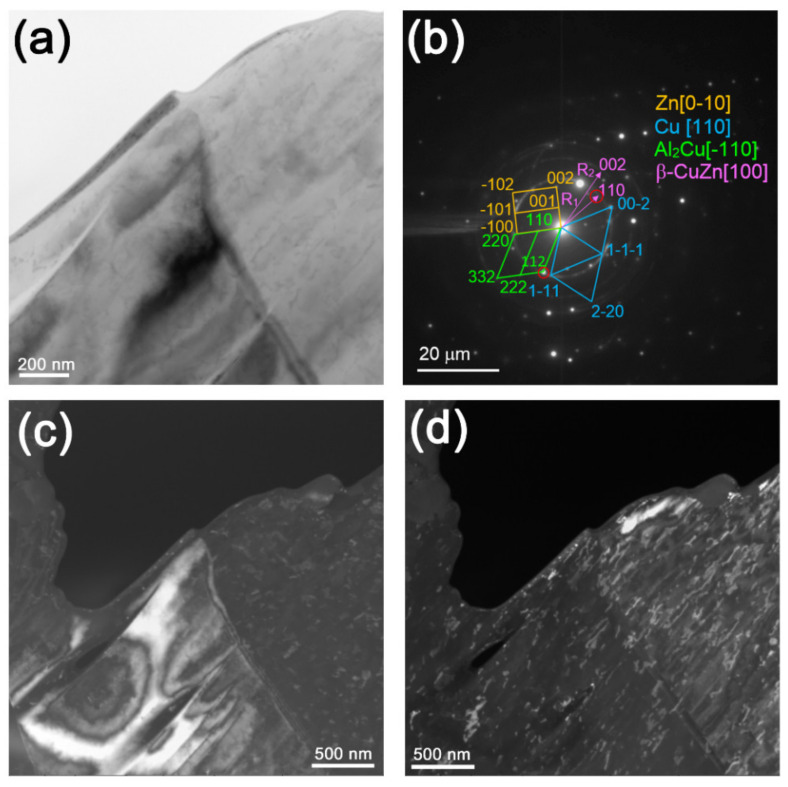
Bright-field image (**a**), SAED pattern (**b**), dark-field image (**c**) obtained using reflections (112)_θ_ and (–110)_CuZn_, and dark-field image (**d**) obtained using reflections (–110)_CuZn_.

**Figure 8 materials-14-05208-f008:**
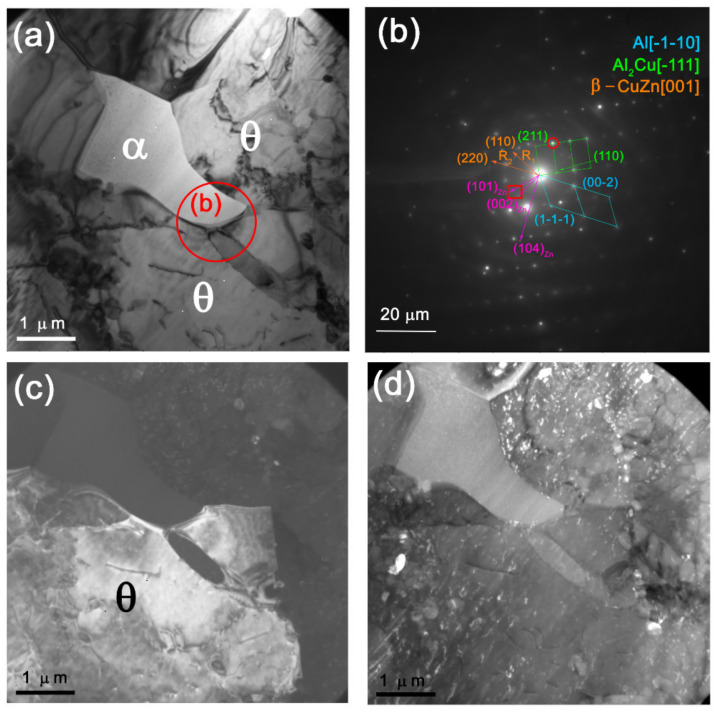
Bright-field image with selected area diaphragm (**a**), SAED pattern (**b**), dark-field image (**c**) obtained using reflections (321)_θ_, and dark-field image (**d**) obtained using reflections (110)_CuZn_ and (002)_Zn_.

**Figure 9 materials-14-05208-f009:**
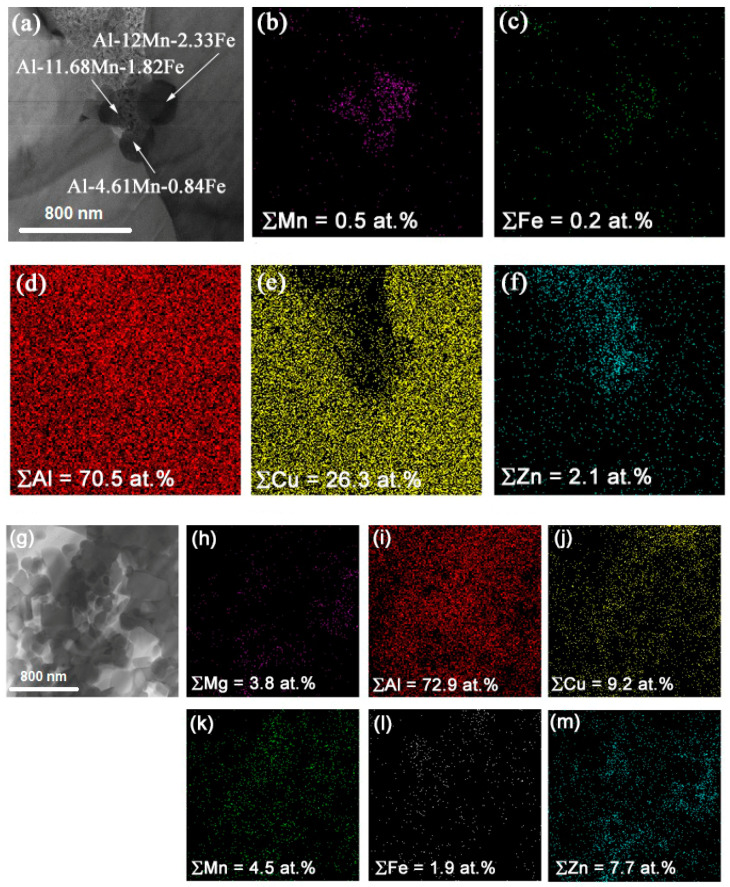
TEM images of Al-Mn-Fe (**a**) and Al-Mg-Fe-Zn (**g**) particles with corresponding EDS element maps (**b**–**f**) and (**h**–**m**), respectively.

**Figure 10 materials-14-05208-f010:**
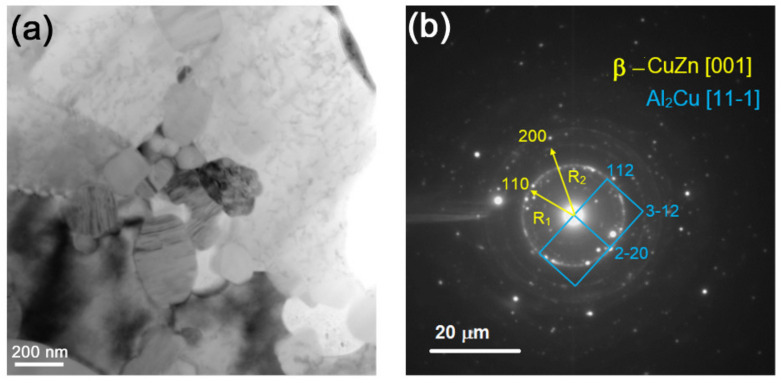
Fine β-CuZn and coarse Al_2_Cu grains (**a**) with corresponding selected area electron diffraction (SAED) pattern (**b**).

**Figure 11 materials-14-05208-f011:**
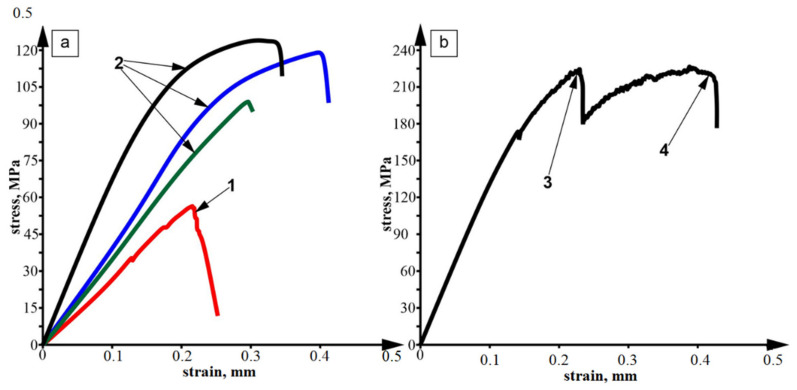
Adhesion (**a**) and tensile (**b**) stress–strain diagrams obtained on the dissimilar lap joints. 1—poor adhesion samples, 2—good adhesion samples, 3—SZ tensile fracture event, 4—substrate fracture event.

**Figure 12 materials-14-05208-f012:**
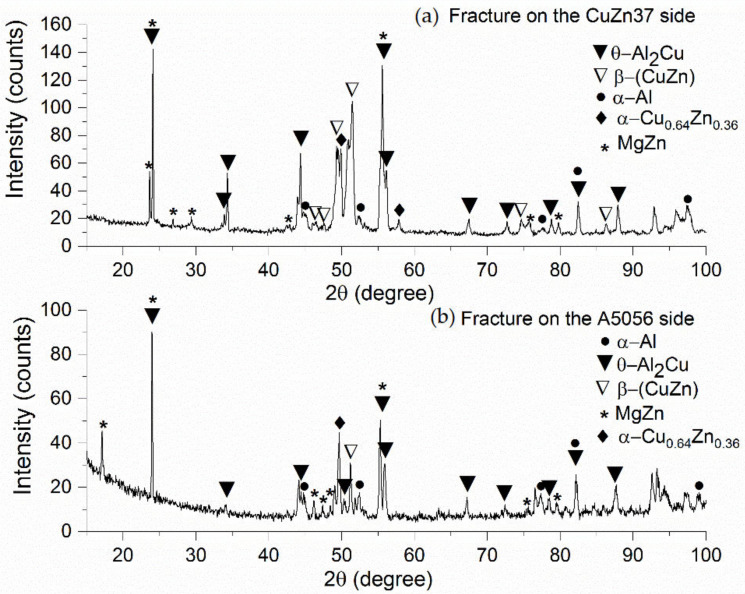
XRD diffractograms of fracture surfaces obtained after adhesion test on CuZn37 (**a**) and AA5056 (**b**).

**Figure 13 materials-14-05208-f013:**
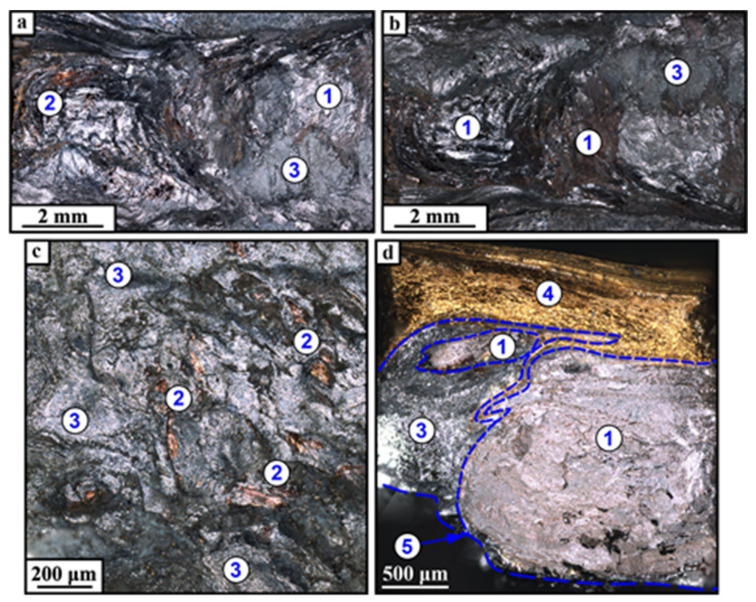
Optical images of normal (**a**–**c**) and tensile (**d**) fracture surfaces. **a**—as seen from the bottom upwards; **b**—as seen from the top downwards; **c**—enlarged fragment of the normal fracture surface.

**Figure 14 materials-14-05208-f014:**
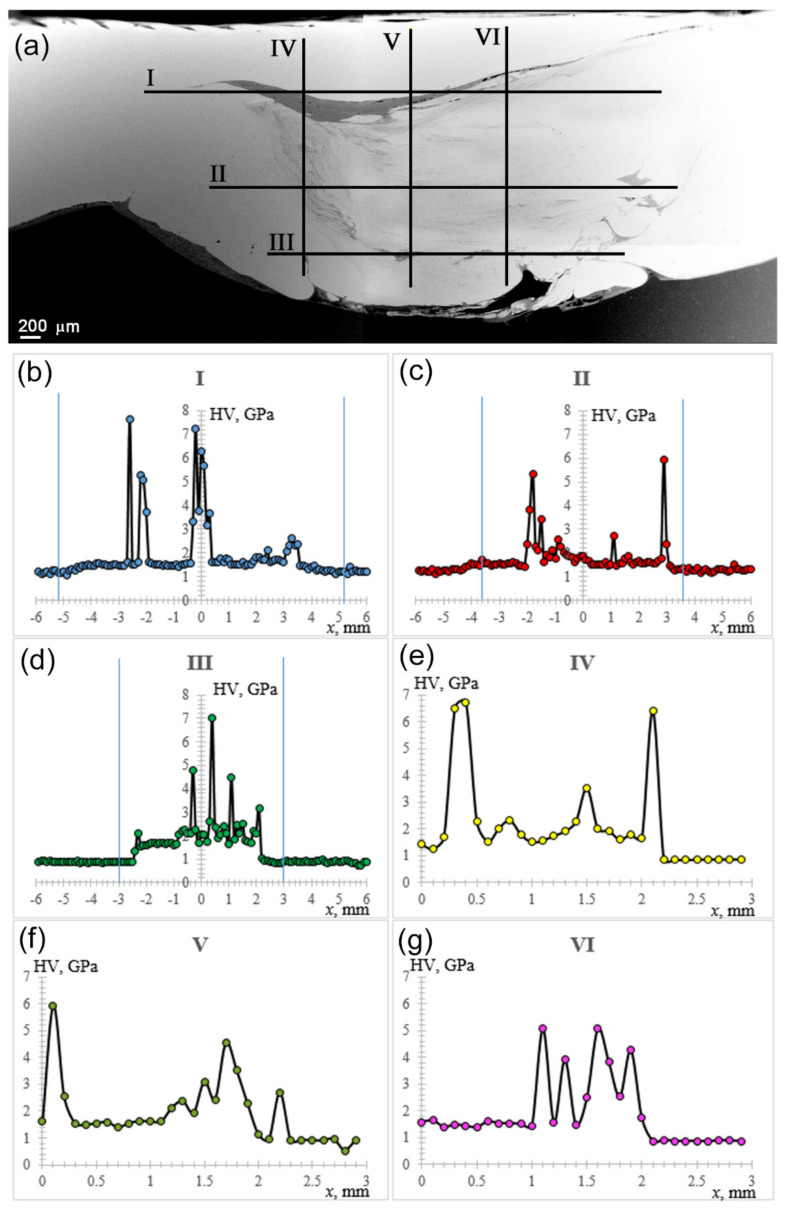
SEM BSE image of the SZ with microhardness measurement lines (**a**), and microhardness profiles obtained along the lines in Figure 14a (**b**–**g**).

**Table 1 materials-14-05208-t001:** Chemical composition of A5056 and CuZn37 brass plates.

Plates		Chemical Element, wt. %.										
	Al	Mg	Fe	Si	Mn	Cu	Zn	Ti	Pb	P	Sb	Be
AA5056	91.9–4.6	4.8–5.8	<0.5	<0.5	0.5–0.8	<0.1	<0.2	<0.02–0.1	-	-		-
CuZn37	-	-	<0.2	-	-	62–65	34.5–38	-	<0.07	0.001	<0.005	<0.002

**Table 2 materials-14-05208-t002:** The FSP parameters used.

Pass Number	Plunge Force, kg	Processing Speed, mm/min	Rotation Rate, rpm
1	1250	100	650
2	1250	100	700

## Data Availability

Data sharing is not applicable to this article.

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
