# Peer review of "Evolution of Microstructure in Friction Stir Processed Dissimilar CuZn37/AA5056 Stir Zone"

_materials, 2021, doi:10.3390/ma14185208_

Round 1
Reviewer 1 Report
The authors of the manuscript show structural results for friction stir welded joints. Work needs improvement according to comments
1. The aim of the study, both in the Abstract and in the Introduction, is lacking!
2. There is no table with the mechanical properties and chemical composition of the materials to be welded
3. The markers in Figure 2a and b should be scaled!
4. Explain why the authors chose such welding parameters? Was there a method for planning the experiment or optimization?
5. Where are the tensile strength results? One should refer to the strength of the parent material
6. The discussion should be supported by Figures and Tables
7. Present 3-4 most important conclusions in the points
8. Technical English requires improvement. The sentences are too complex!
Author Response
The authors of the manuscript show structural results for friction stir welded joints. Work needs improvement according to comments
- The aim of the study, both in the Abstract and in the Introduction, is lacking!
A: Thank you. Corrected to read: «The objective of this research is, therefore, to study the microstructural evolution of SZ in dissimilar lap FSP(FSW) of a Cu-Zn alloy to an Al-Mg alloy with special attention given to formation of primary Al/Cu and secondary Mg/Zn IMCs”
- There is no table with the mechanical properties and chemical composition of the materials to be welded
A: Table 1 containing the alloy compositions was added to the Materials and Methods section as well as ultimate strength values.
- The markers in Figure 2a and b should be scaled!
A: Corrected
- Explain why the authors chose such welding parameters? Was there a method for planning the experiment or optimization?
A: The FSP parameters were chosen basing upon our earlier experiments on FSP of Cu/Al
- Where are the tensile strength results? One should refer to the strength of the parent material
A: Ultimate strength limits of the parents metals have been added to the Materials and Methods section. In fact, tensile strength of the composite FSPed SZ was close to that of the AA50506 substrate.
- The discussion should be supported by Figures and Tables
A: We did our best to use Figures in Discussion section
- Present 3-4 most important conclusions in the points
A: Done
- Technical English requires improvement. The sentences are too complex!
A: Corrected. Some sentences were rewritten for more clarity
Reviewer 2 Report
Congratulations on the work. This work aims to study the microstructural evolution of the stir zone in different Friction stir processing/welding (FSP/FSW) from a Cu-Zn alloy to an Al-Mg alloy. In general, the article is written in a succinct way, has experimental support and the language is simple, however, I would like to leave some suggestions for improvements.
The Introduction could be improved, although providing a slight review of recent developments it should be more critical and comprehensive. The Introduction section is expected to have an extensive literature review followed by a deep and critical analysis of the state of the art.
Line 65: Please clarify the meaning of EDM (Electric Discharge Machining), what do you mean is the samples were Wire Cut Electric Discharge Machining (WCEDM)?
Line 70: the authors state that they used the SEM equipment, they must write its meaning in full (scanning electron microscope) clarify the model and country as described for example in Lines 66 and 67 for the microscope.
Please check throughout the article if all acronyms used were written in full. You can only write an acronym after having written its meaning at least once. Other example where this situation happens: XRD (line 73).
I suggest you put the caption on the image and remove it from the description in figure 1. this way it improves the reader's understanding.
In the results, figure 3, the authors placed figure 4 (c) and (f), however, they do not comment on the graphics in the text. please describe what you intend to show in the text with these images. Please rectify these situations.
I suggest that the authors put the reference of each figure with the same size and font, along with the images its reference varies. In some images they appear below the images in other situations they appear on top of the images. Another situation that concerns me is related to the figures is the fact that they appear for example Figure 6 (e) you have six figures with the reference (e). These situations can make it difficult for the reader to understand.
I suggest you put at least 15 to 20 more references. It should complement the results and discussion to support the results obtained. The authors don’t mention other works search in the results and discussion section, it would be interesting to show future trends.
The conclusion also needs to be extended. Please extend with a comparison with similar studies.
Author Response
Congratulations on the work. This work aims to study the microstructural evolution of the stir zone in different Friction stir processing/welding (FSP/FSW) from a Cu-Zn alloy to an Al-Mg alloy. In general, the article is written in a succinct way, has experimental support and the language is simple, however, I would like to leave some suggestions for improvements.
The Introduction could be improved, although providing a slight review of recent developments it should be more critical and comprehensive. The Introduction section is expected to have an extensive literature review followed by a deep and critical analysis of the state of the art.
A: More references and analysis were added to the Introduction
Line 65: Please clarify the meaning of EDM (Electric Discharge Machining), what do you mean is the samples were Wire Cut Electric Discharge Machining (WCEDM)?
A: The sentence was revised to read: “Samples for microstructural and mechanical characterization were cut off the FSPed tracks using a wire-cut electric discharge machining (WCEDM) (Figure 1).”
Line 70: the authors state that they used the SEM equipment, they must write its meaning in full (scanning electron microscope) clarify the model and country as described for example in Lines 66 and 67 for the microscope.
A: The sentence was revised to read:” Chemical element distributions were examined using an EDS attachment to the above-indicated scanning electron microscope.”
Please check throughout the article if all acronyms used were written in full. You can only write an acronym after having written its meaning at least once. Other example where this situation happens: XRD (line 73).
A: The sentence was revised to read:” Phase composition of the SZ was determined using an X-ray diffractometer XRD-7000S (Shimadzu, Kyoto, Japan), .radiation
I suggest you put the caption on the image and remove it from the description in figure 1. this way it improves the reader's understanding.
A:Done.
In the results, figure 3, the authors placed figure 4 (c) and (f), however, they do not comment on the graphics in the text. please describe what you intend to show in the text with these images. Please rectify these situations.
A: A sentence describing the Figs. 3c , f was added as follows:” The EDS profiles (Fig.3c, f) were obtained along the yellow lines shown in Fig.3a and Fig.3e which allowed observing element distributions across Al2Cu particles and -Al(Cu,Zn) spaces”.
I suggest that the authors put the reference of each figure with the same size and font, along with the images its reference varies. In some images they appear below the images in other situations they appear on top of the images. Another situation that concerns me is related to the figures is the fact that they appear for example Figure 6 (e) you have six figures with the reference (e). These situations can make it difficult for the reader to understand.
A:Corrected
I suggest you put at least 15 to 20 more references. It should complement the results and discussion to support the results obtained. The authors don’t mention other works search in the results and discussion section, it would be interesting to show future trends.
A:Corrected
The conclusion also needs to be extended. Please extend with a comparison with similar studies.
A: Corrected
Reviewer 3 Report
In paragraph 2, Materials and Methods, the method, apparatus and sample for measuring microhardness should be presented.
The composition of the eutectic formed at the bottom of the SZ must be analyzed more carefully.
The eutectic is formed in the area of the AlMg5 alloy plate, and the EDS analysis shows a high Al content. Therefore it could be a ternary eutectic.
It is surprising the high content of Zn in the eutectic area. It should be explained.
Figure 5a does not indicate pos.1.
The appearance of the s Al-Mn-Fe and Al-Mg-Fe precipitates should be explained.
The conclusions should also refer to the influence of the number of passes on the microstructural evolution of the SZ.
Author Response
In paragraph 2, Materials and Methods, the method, apparatus and sample for measuring microhardness should be presented.
A: Corrected
The composition of the eutectic formed at the bottom of the SZ must be analyzed more carefully.
The eutectic is formed in the area of the AlMg5 alloy plate, and the EDS analysis shows a high Al content. Therefore it could be a ternary eutectic.
A: An EDS map in Fig.3b shows total aluminum content 41.2 at.% mainly from residual aluminum alloy grains on the periphery, while it is very low in the eutectic region. Also XRD from the fracture surfaces shows the presence of MgZn. However, we agree that some low concentration of Al atoms is feasible in the MgZn and there could be a ternary eutectics.
It is surprising the high content of Zn in the eutectic area. It should be explained.
A: All zinc comes from CuZn37 brass and concentrates in spaces between the Al2Cu IMCs as a melt. Only 0.05 % Al can be dissolved in Zn at 20°C. All explanations were added to the text.
Figure 5a does not indicate pos.1.
A: Thank you. The sentence has been revised to read: “These layered structures are composed of oversaturated solid solutions (Figure 5c-e, pos.1) and IMCs (Figure 5a-e, pos.2).
The appearance of the s Al-Mn-Fe and Al-Mg-Fe precipitates should be explained.
A: The presence of Fe-enriched insoluble particles is common with the aluminum alloys. In addition, iron contamination may be the result of steel FSP tool diffusion wear.
The conclusions should also refer to the influence of the number of passes on the microstructural evolution of the SZ.
A: All experiments were carried out only on 2-pass FSPed samples since 1-pass FSP resulted in formation of a discontinuity.
Reviewer 4 Report
The study “Evolution of microstructure in friction stir processed dissimilar CuZn37/AA5056 stir zone” is an interesting peace of work devoted to a promising area of friction stir processing. The study involves multiple modern techniques and methods. The obtained results are of a reliable and well presented.
The article consists of interesting experiments; however, none of them is discussed, but is presented as a fact. At the beginning of paper there was an attempt to compare two passes, but there is no comparison further in the text. It is also unclear why only two passes were made. The structure and properties of the FSW are described, but there is no comparison with the alloy without FSW. In the article, it is necessary to write the discussion not only at the end, but also in the full text. The conclusions in the article are enough presented, it is better to expand the conclusions and number them.
There are several flaws that should be corrected and justified before publishing in Materials. The comments and advice are drawn as follows:
- Line 28: The description of IMC abbreviation is missed
- Lines 38-39: “Dissimilar Cu-Al welded joints always were of practical interest and therefore attempts were undertaken to replace copper for a copper alloy such as brass.” What was the reason of replacement? Should be clarified.
- Line 50: “according to both solid state and liquation mechanisms” what mechanism do authors mean? The reference is required.
- It is not supported by the introduction why the brass of such composition was used. Is it previously reported that the Zn addition in Cu improves the weldability or properties of joint? This point should be strengthened in introduction.
- The chemical compositions of the studied alloys must be provided in Materials and Methods section
- Why was the rotation rate increased at the second pass?
- Figure 3. The pass number should be given in the caption. The lines EDS profiles (c,f) should be presented in (b,e). The revealed phases also should be indicated in the images.
- The cavitation is observed in Figure 3 (a, d) and Figure 5 (a). It is not mentioned in the text.
- The eutectic colonies are observed in microstructure. It raises the question in what atmosphere (argon, air) the welding was carried out. It should be mentioned in Methods section. The liquid phase is prone to oxidizing that results in increased brittleness of joints. In this case I also recommend adding O element in further EDS maps.
- Figure 5. sign (a) missed. The XRD spectrum image (b) and all labels in Figure are too small. The figure should be improved.
- The numbers for points in Figure 6a are too small and cannot be identified. These areas (c) should be marked-up in (a)
- Lines 180-185 contain Fe, it is not clear. Please, explain this data.
- Figure 11 b. The comparing of stress-strain diagram of dissimilar lap joints with that for basic alloys CuZn37 and AA5056
Author Response
The study “Evolution of microstructure in friction stir processed dissimilar CuZn37/AA5056 stir zone” is an interesting peace of work devoted to a promising area of friction stir processing. The study involves multiple modern techniques and methods. The obtained results are of a reliable and well presented.
A: Thank you
The article consists of interesting experiments; however, none of them is discussed, but is presented as a fact. At the beginning of paper there was an attempt to compare two passes, but there is no comparison further in the text. It is also unclear why only two passes were made. The structure and properties of the FSW are described, but there is no comparison with the alloy without FSW. In the article, it is necessary to write the discussion not only at the end, but also in the full text. The conclusions in the article are enough presented, it is better to expand the conclusions and number them.
A:Corrected
There are several flaws that should be corrected and justified before publishing in Materials. The comments and advice are drawn as follows:
Line 28: The description of IMC abbreviation is missed
A: Abbreviation such as IMC was introduced above with the Abstract
Lines 38-39: “Dissimilar Cu-Al welded joints always were of practical interest and therefore attempts were undertaken to replace copper for a copper alloy such as brass.” What was the reason of replacement? Should be clarified.
A: It is of course the reduction of copper content and saving costs but maintaining the characteristics at some acceptable level.
Line 50: “according to both solid state and liquation mechanisms” what mechanism do authors mean? The reference is required.
A: The sentence was revised to read: “However, all alloys contain numerous other alloying elements which also could react during FSP and thus have effect on the formation of SZ according to solid state diffusion or solidification from a melt.
It is not supported by the introduction why the brass of such composition was used. Is it previously reported that the Zn addition in Cu improves the weldability or properties of joint? This point should be strengthened in introduction.
A: Zinc is used as a barrier to retard the IMC growth in dissimilar FSW of copper to aluminum and thus serves to improve mechanical strength of the dissimilar joint. Our purpose was to use it for improving strength of an in-situ FSP composite. The Introduction section was revised to include that.
The chemical compositions of the studied alloys must be provided in Materials and Methods section
A: Thank you. Table 1 contaiing the compositions of both alloys was added to the Section. Also tensile strengths were added.
Why was the rotation rate increased at the second pass?
A: Somewhat increased rotation rate was used to increase heat input and thus improve plasticization of the IMCs formed after the first pass.
Figure 3. The pass number should be given in the caption. The lines EDS profiles (c,f) should be presented in (b,e). The revealed phases also should be indicated in the images.
A: All the results relate only to the 2-pass FSPed samples.
The cavitation is observed in Figure 3 (a, d) and Figure 5 (a). It is not mentioned in the text.
A: No cavitation was observed after 2-pass FSP
The eutectic colonies are observed in microstructure. It raises the question in what atmosphere (argon, air) the welding was carried out. It should be mentioned in Methods section. The liquid phase is prone to oxidizing that results in increased brittleness of joints. In this case I also recommend adding O element in further EDS maps.
A: No gas shielding was applied during the FSP. Some oxidizing always occurs during the FSP but it relates only to aluminum which is the most actively oxidizing element. The stirring occurs inside the metal so that not much oxygen access is hindered. In fact, no oxygen was not detected during the EDS examination.
Figure 5. sign (a) missed. The XRD spectrum image (b) and all labels in Figure are too small. The figure should be improved.
A: Corrected
The numbers for points in Figure 6a are too small and cannot be identified. These areas (c) should be marked-up in (a)
A: Corrected
Lines 180-185 contain Fe, it is not clear. Please, explain this data.
A: The presence of Fe-enriched insoluble particles is common with the aluminum alloys (see Table 1). In addition iron contamination may be the result of steel FSP tool diffusion wear.
Figure 11 b. The comparing of stress-strain diagram of dissimilar lap joints with that for basic alloys CuZn37 and AA5056
A: Ultimate strength limits of the parents metals have been added to the Materials and Methods section. In fact, tensile strength of the composite FSPed SZ was close to that of the AA50506 substrate.
Round 2
Reviewer 1 Report
The authors revised the manuscript according to comments.It is now ready for publication in Materials
Reviewer 2 Report
I would like to congratulate the authors for the improvement made.
I leave only one note that is just a distraction: Please correct the following sentence: line 150: "The end part of the profile in Figure 3f s show unreacted CuZn37 (Figure 3e)."
Reviewer 4 Report
The authors responded to all comments, revised the manuscript and revised figures. I believe the article can be published without further changes.